# Physiological and Morphological Response Mechanisms of *Theobroma cacao* L. Rootstocks Under Flooding and Evaluation of Their Adaptability

**DOI:** 10.3390/plants15010122

**Published:** 2026-01-01

**Authors:** Maria Luiza Pereira Barbosa Pinto, Vinicius de Souza Oliveira, Jeane Crasque, Basílio Cerri Neto, Thayanne Rangel Ferreira, Carlos Alberto Spaggiari Souza, Antelmo Ralph Falqueto, Thiago Corrêa de Souza, José Altino Machado Filho, Lúcio de Oliveira Arantes, Carla da Silva Dias, Enilton Nascimento de Santana, Karin Tesch Kuhlcamp, Sara Dousseau-Arantes

**Affiliations:** 1Departamento de Ciências Agrárias e Biológicas, Universidade Federal do Espírito Santo, Centro Universitário Norte do Espírito Santo, São Mateus 29932-540, Espírito Santo, Brazil; luiza_14p@hotmail.com (M.L.P.B.P.); antelmofalqueto@gmail.com (A.R.F.); 2Instituto Capixaba de Pesquisa, Assistência Técnica e Extensão Rural—Centro Regional de Desenvolvimento Rural—Norte, Linhares 29901-443, Espírito Santo, Brazil; altino@incaper.es.gov.br (J.A.M.F.); lucio.arantes@incaper.es.gov.br (L.d.O.A.); carla.dias@incaper.es.gov.br (C.d.S.D.); enilton@incaper.es.gov.br (E.N.d.S.); karin.kuhlcamp@incaper.es.gov.br (K.T.K.); 3Centro de Ciências Humanas e Naturais, Universidade Federal do Espírito Santo, Centro de Ciências Humanas e Naturais, Vitória 29075-910, Espírito Santo, Brazil; jeanecrasker@gmail.com (J.C.); basiliocerri@yahoo.com.br (B.C.N.); thayannerangel85@gmail.com (T.R.F.); 4Comissão Executiva do Plano da Lavoura Cacaueira, Linhares 29900-000, Espírito Santo, Brazil; spaggiari.ceplac@gmail.com; 5Instituto de Ciências da Natureza (ICN), Universidade Federal de Alfenas, Alfenas 37130-000, Minas Gerais, Brazil; thiagonepre@hotmail.com

**Keywords:** carbohydrate, environmental stress, gas exchange, lenticel

## Abstract

The response of cocoa (*Theobroma cacao* L.) to low oxygen availability in the soil and the possibility of recovery after stress relief are associated with the plasticity capacity of each genotype; however, studies evaluating the influence of rootstock on stress response are scarce. Thus, in the northern region of the state of Espírito Santo, municipality of São Mateus, the physiological, biochemical, and anatomical responses and recovery capacity of cocoa PS-1319 grafted onto the rootstocks TSH-1188, Cepec-2002, Pará, Esfip-02, and SJ-02 were evaluated under flooded conditions. The plants were subjected to flooding for 60 days, and their recovery capacity was assessed after this period. The gas exchange, relative chlorophyll content, stem and leaf anatomy, photosynthetic pigments, and carbohydrates were evaluated. All genotypes showed reductions in net photosynthetic assimilation, stomatal conductance, and transpiration rate in the flooded environment compared to the non-flooded environment. All pigments were degraded, with average values of Chl *a*, Chl *b*, total Chl, and total carotenoids of 9.33, 10.418, 19.75, and 590.75 μg.mL^−1^ in the non-flooded environment and 6.43, 7.69, 14.12, and 500.33 μg.mL^−1^ in the flooded environment. The rootstocks Cepec-2002 and Esfip-02 showed the highest carotenoid accumulation, with 585.78 and 650.47 μg.mL^−1^, respectively, when compared to SJ-02 (474.03 μg.mL^−1^), Pará (491.58 μg.mL^−1^), and TSH-1188 (525.86 491.58 μg.mL^−1^). The Pará rootstock did not show differences in stomatal density between environments, with values of 32.25 in flooding, 34.83 in non-flooding, and 31.61 in recovery. During flooding, lenticels formed in all rootstocks. After recovery, all rootstocks normalized their gas exchange, carbohydrate levels, and anatomy, showing that the root system was able to re-establish its functions, making these rootstocks suitable for areas at risk of flooding.

## 1. Introduction

The cocoa (*Theobroma cacao* L.) is one of the most important tropical fruit trees, mainly due to the commercial value of its almonds, which are the raw material for the chocolate industry [1]. In 2023, the world produced approximately 5,600,100.42 t of almonds, with emphasis on the Ivory Coast (2,377,442.05 t), Ghana (653,700 t), Indonesia (641,741.02 t), Ecuador (375,719.14 t), and Brazil (296,145 t). Together, these countries are responsible for more than 77% of the world’s production of cocoa almonds [2].

One of the most limiting factors in the production of cocoa crops is water availability. For crops to develop well, annual rainfall of 1500 to 2000 mm is required, and must be well distributed [3]. Traditionally, the main commercial cocoa-producing plantations are located in regions where water demand is satisfactorily met by the crop [4]. However, climate change has been one of the obstacles to agricultural production [1]. One of the changes observed is the rainfall regime, which, in excess, can reduce the soil’s drainage capacity [5], in addition to causing flooding in riverside regions, causing waterlogging of cultivated soils.

Soil flooding can result in a reduction in gas concentration, resulting in the low availability (hypoxia) or complete lack (anoxia) of oxygen [6], causing metabolic and physiological disorders in plants [7], such as the production of free radicals and oxidative stress [8], leading to cell lysis, in addition to a reduction in the synthesis of photosynthetic pigments [9], decreased gas exchange [10], a drop in the photosynthetic rate [11], stomatal closure [12], and loss in the translocation of carbohydrates from leaves to roots, thus producing an accumulation of photosynthates in the leaves [13]. All of these factors can limit plant development and growth, as well as reduce productivity.

It is important to emphasize that the ability of plants to develop normally after periods of flooding is fundamental to ensuring their survival in the field; however, the responses of different species and genotypes to these conditions are still limited [8]. Thus, the evaluation of different cocoa genotypes during periods of flooding and after stress plays an important role in understanding the ability of plants to restore their functions after flooding, which can be useful in decision-making, guiding farmers in cultural practices and correct crop management [14].

Although scarce, some studies have investigated the recovery of cocoa plants under post-flooding conditions. Sena Gomes and Kozlowski [15] verified the partial recovery of the Catongo genotype 11 days after 30 days of flooding, with the main recovery observed in the root system. Almeida et al. [16] found that cocoa hybrids flooded for 35 days had a recovery of between 78 and 100% of their photosynthetic capacity 10 days after the stressful conditions. Pinto et al. [14] found that cocoa plants had compromised photosynthetic apparatus during flooding; however, after the humidity conditions were normalized, the plants showed signs of recovery.

It is worth noting that a large part of commercial cocoa cultivation is established with seedlings from grafting, where the effects of scion and rootstock combinations are not known. The cocoa genotypes Pará, TSH-1188, Esfip-02, Cepec-2002, and SJ-02 are commonly used as rootstocks due to their ability to help the grafted plant overcome soil and disease challenges. These genotypes are selected for crucial characteristics such as vegetative vigor, which ensures rapid initial seedling growth, and resistance to pathogens like witches’ broom (*Moniliophthora perniciosa*) and Ceratocystis wilt (*Ceratocystis cacaofunesta*), which pose significant threats to cocoa cultivation. However, despite the notable importance of these genotypes in seedling production, providing robustness and quality to the plants, they are not characterized in terms of tolerance to flooded environments. Thus, understanding the effects of flooding on the development of the rootstocks TSH-1188, Cepec-2002, Pará, Esfip-02, and SJ-02, which are used in cocoa production and are of global importance in the chocolate manufacturing industry, is essential to obtain information and identify rootstocks that are more tolerant to flooding, ensuring a longer lifespan and greater sustainability for cocoa plantations in flooded areas [14].

Therefore, studies that seek to understand the effects of flooding on different plant species are important to understand the best management methods, in addition to selecting genotypes that can serve as an alternative to minimize the damage caused by stress. Thus, the objective of this study was to evaluate the physiological, biochemical, and anatomical changes in response to flooding and the recovery capacity of *T. cacao* L. rootstocks.

## 2. Materials and Methods

### 2.1. Experiment Conduction and Seedling Production

The study was conducted for 90 days, during the months of May to July, at the Capixaba Institute for Research, Technical Assistance and Rural Extension (INCAPER), located in the municipality of Linhares, Espírito Santo, at the geographic coordinates: 19°25′00.1″ S and 40°04′35.3″ W. The climate data during the experimental period were monitored through an automatic meteorological station (Figure 1).

The cocoa (*Theobroma cacao* L.) seedlings used were produced at the Filogônio Peixoto Experimental Station, belonging to the Cocoa Research Center, a research body of the Executive Committee of the Cocoa Cultivar Plan (CEPLAC), located in the municipality of Linhares/ES. The genotypes used were PS-1319 scions, with the rootstock genotypes TSH-1188, Cepec-2002, Pará, Esfip-02, and SJ-02. The seedlings were produced from propagules obtained from mother plants supplied by the CEPLAC Active Germplasm Bank. The genealogy and agronomic descriptors of the six cocoa genotypes used in the experiment are described in Table 1. The cleft grafting technique was used, performed 5 months after sowing the rootstock [17]. Two months after grafting, the seedlings were transplanted into polyethylene bags (25 × 35 cm), one plant per bag, containing a sandy loam substrate with the following chemical characteristics: 0.008 kg m^−3^ of phosphorus, 0.024 kg m^−3^ of potassium, 0.007 kg m^−3^ of sulfur, 0.222 kg m^−3^ of iron, 0.0026 kg m^−3^ of zinc, 0.0006 kg m^−3^ of copper, 0.015 kg m^−3^ of manganese, 0.00024 kg m^−3^ of boron, 0.005 kg m^−3^ of sodium, 0.0486 kg m^−3^ of magnesium, 0.44088 kg m^−3^ of calcium, and 9 kg m^−3^ of organic matter.

The plants were transferred to masonry tanks lined with white tarpaulin to prevent water infiltration (Figure 2). A shade cloth was installed 3 m high to block 50% of incident radiation, where the seedlings were acclimatized for 60 days before being subjected to treatments. During acclimatization, the seedlings were irrigated daily with a sprinkler system for 45 min every 4 h. Foliar nitrogen fertilization and pest control were carried out when necessary.

At nine months of age, the plants were subjected to flooding for 60 days, maintaining the water level up to the base of the stem. Water was replenished as needed to prevent column contraction. The dissolved oxygen content was maintained at approximately 0.00898 kg m^−3^, these values being measured by averaging three readings taken after 3 days without water replenishment. After 60 days of flooding, the tank was emptied, and the plants were irrigated for 30 days under the same conditions as the control, to evaluate the recovery capacity of the evaluated genotypes.

### 2.2. Gas Exchanges and Chlorophyll Relative Content

Gas exchange and relative chlorophyll content (SPAD) were assessed at 5, 7, 9, 11, 13, 17, 19, 24, 29, 41, 48, and 55 days after flooding, and during recovery at 7, 14, 21, and 29 days after flooding was suspended. Two plants per replicate were used between 7:00 and 10:00 a.m., in fully expanded leaves in the median portion of the plant, at the third node of one leaf per plant. Gas exchange was performed using an infrared gas analyzer (IRGA), model LI-6400 (LI-COR, Lincoln, NE, USA), with a saturating irradiance of 1000 [μmol(photon) m^−2^ s^−1^] photons, a temperature of 25 °C, and a CO_2_ of 400 ppm. From these data, the following were calculated: the ratio between intra/extracellular CO_2_ concentrations (*C*_i_/*C*_a_); net photosynthetic rate (*P*_N_) [μmol(CO_2_) m^−2^ s^−1^]; stomatal conductance (*g*_s_) [mmol(H_2_O) m^−2^ s^−1^]; transpiration rate (*E*) [mmol(H_2_O) m^−2^ s^−1^]; and intrinsic water-use efficiency (WUE) [mol(CO_2_) mol(H_2_O)^−1^]. A portable chlorophyll meter (SPAD-502, Minolta^®^, Kantō-chihō, Tokyo, Japan) was used to determine the relative chlorophyll content.

### 2.3. Stem and Leaf Anatomical Evaluations

For anatomical evaluations, one plant per block was used. Lenticel formation was qualitatively assessed by visual evaluation. Stem fixation was performed in FAA solution (formalin (5%), acetic acid (5%), and ethyl alcohol (90%) in 100 mL) at 70% for fifteen days, and then transferred to 70% alcohol, and leaves were fixed directly in 70% alcohol. Histological sections according to [18] were obtained freehand with the aid of a stainless steel blade. Stem and leaf cross sections were subjected to clarification with sodium hypochlorite solution and then washed in distilled water until excess hypochlorite was removed. Sections were stained with safrablau (safranin + Astra blue) and kept in distilled water; paradermal sections of leaves were made using the epidermal impression technique. The temporary slides were observed under a bright-field microscope (Euromex, Gelderland, Duiven, The Netherlands). The images were captured with a microcamera (CMEX 5, Euromex, Gelderland, Duiven, The Netherlands), and the biometric measurements of the tissues were performed with the ImageFocus 4 software. The parameters evaluated for the leaf were stomatal density [mm^2^], polar diameter/equatorial diameter (PD/DE) ratio, and for the stem, the lenticel length [mm] and lenticel height [mm].

### 2.4. Photosynthetic Pigments

Pigment extraction was performed 60 days after flooding. Chlorophyll and carotenoid contents were determined using 1–2 cm fragments from the median region of the leaf blade, discarding the central vein, from completely expanded and fresh leaves, from the third node below the apex of the plants, chosen at random. The extraction and quantification of chlorophyll *a*, *b*, and total were performed according to the methodology of Arnon [19]. The extraction and quantification of carotenoids were performed according to the methodology described by Duke and Kenyon [20], using the molar absorptivity coefficients of Sandmann et al. [21].

### 2.5. Carbohydrate Extraction and Quantification

After 60 days of flooding and 30 days after the stress was suspended, leaves and roots of the plants were collected to quantify the levels of total soluble sugars (TSS), reducing sugars (RS), and starch. The plant material was dried in an air circulation and renewal oven (TE-394/3, Tecnal, Piracicaba, São Paulo, Brazil) at 70 °C until constant weight, ground in a Willye-type mill (TE-650, Tecnal and brand) with a 0.6 mm mesh 30 sieve (TE-650C, Tecnal, Piracicaba, São Paulo, Brazil), and stored in a freezer at −18 °C (CHA22, Consul, Santa Catarina, Brazil) until extracts were obtained. The extracts were obtained according to [22], using 0.2 g of leaves and 0.3 g of roots. For the quantification of total soluble sugars (TSS) and starch, the Anthrone method [23] was used, with modifications, using 2 mL of Anthrone solution at 0.19% in sulfuric acid at 93.33%, in a reaction volume of 3 mL, subjected to 100 °C for 3 min. Reducing sugars (RS) were quantified according to the protocol described by [24], through the Dinitrosalicylic Acid (DNS) method.

### 2.6. Statistical Analysis

The treatments were arranged in a randomized block experimental design, in a factorial scheme (5 × 2). The first factor consisted of five different cocoa rootstocks (TSH-1188, Cepec-2002, Pará, Esfip-02, and SJ-02). The second factor consisted of two conditions, namely flooded and non-flooded. Four blocks were evaluated, each plot consisting of ten plants. Two plants per block were used to measure gas exchange and relative chlorophyll content (SPAD), photosynthetic pigments, and carbohydrate content. For anatomical evaluations of the stem and leaf, one plant per block was used.

The data were subjected to analysis of variance by the F-test at 5% probability. Subsequently, the means were compared by the Scott–Knott clustering test at a significance level of 5% and by the Dunnett test. Statistical analyses were performed using the SISVAR [25] and R programs [26].

## 3. Results

### 3.1. The Recovery of Gas Exchanges and SPAD

All genotypes cocoa (*Theobroma cacao* L.) showed decreases in net photosynthetic rate (*P*_N_) (Figure 3), stomatal conductance (*g*_s_) (Figure 3), transpiration rate (*E*) (Figure 4), intrinsic water-use efficiency (WUE) (Figure 4), and relative chlorophyll content (SPAD unit) in leaves (Figure 5). However, there was an increase in the ratio of intra/extracellular CO_2_ concentrations (*C*_i_/*C*_a_) (Figure 4) under flooding compared to the non-flooded environment. The Esfip-02 genotype showed more sensitive behavior at the beginning of flooding (Figure 3 and Figure 6), while SJ-02 was more stable, since its behavior in the flooded environment did not differ from the behavior during the non-flooded environment when compared to the other genotypes studied. All genotypes, after 40 days of flooding, showed stability in the parameters, maintaining photosynthesis below 2 [μmol(CO_2_) m^−2^ s^−1^], conductance below 0.1 [(H_2_O) m^−2^ s^−1^], and transpiration below 1.0 [(H_2_O) m^−2^ s^−1^] (Figure 3).

After recovery, all genotypes restored their photosynthesis (Figure 6), transpiration (Figure 6), carbon concentration ratio (Figure 7), and WUE (Figure 8). However, at 29 days of recovery, the genotypes Cepec-2002, Esfip-02, and SJ-02 reduced internal carbon (Figure 7). The genotypes TSH-1188 and Pará were unable to return stomatal conductance to the non-flooded condition level after only 30 days of recovery (Figure 6), while the genotypes Pará and Cepec-2002 were unable to recover their SPAD (Figure 8).

### 3.2. Stem and Leaf Anatomy

From the ninth day of flooding, all plants presented hypertrophied lenticels, which remained until the end of the experimental period (Figure 9), and after the flooding was removed, the stem began to reduce the openings. The length of the lenticels (Figure 10) was statistically equal in the genotypes under flooding, and only SJ-02 and Pará did not reduce the length of the lenticels after recovery (Table 2), but there was a reduction in the height of the lenticels after the suspension of the stress (Table 2). The stomatal density increased for SJ-02 and Esfip-02 under flooding, while only SJ-02 reduced the DP/DE ratio in flooding when compared to the other environments (Table 2).

### 3.3. Flooding on Pigment Contents

There was no statistical difference between the genotypes for chlorophyll content, which was only affected by the environment. Flooding caused a reduction in the content of photosynthetic pigments (Table 3). Carotenoid content differed among the genotypes studied, with Esfip-02 and Cepec-2002 presenting the highest values among the five rootstocks evaluated (Table 3). Among the environments, carotenoid content was degraded by flooding (Table 3).

### 3.4. Effect of Flooding and Recovery on the Carbohydrates

All genotypes showed an accumulation of reducing sugar and total soluble sugar in the leaves during flooding. After flooding was suspended, only SJ-02 did not reduce the accumulation of reducing sugar in the leaves. Except for Pará, soluble sugar in the leaves was reduced during flooding suspension for the other genotypes (Table 4). During flooding, TSH-1188 accumulated more reducing sugar and less soluble sugar; on the other hand, Esfip-02 and Cepec-2002 showed the opposite behavior (Table 4). In the root system, the accumulation of total soluble sugar increased with flooding, while starch content increased during recovery, and the genotypes Esfip-02, SJ-02, and TSH-1188 accumulated greater amounts of starch in the roots. In the leaves, flooding caused starch accumulation, which reduced after recovery (Table 4).

## 4. Discussion

### 4.1. Gas Exchange and SPAD

Cocoa tree (*Theobroma cacao* L.) plants had distinct behaviors in relation to the different genotypes and flooded, non-flooded, and control environments to which they were subjected. When kept under flooding, the plants showed negative behavior in relation to gas exchange (Figure 3, Figure 4 and Figure 5). The reduction in net photosynthetic rate under flooding may be related to stomatal closure by photoinhibition [27] or to photochemical impairment, causing limitation in leaf gas exchange [28]. The rootstock SJ-02 showed low photosynthetic rates in the control plants. However, due to the low photosynthetic rates in relation to the other rootstocks, the effect of flooding was not very pronounced in SJ-02. This finding is consistent with Bertolde et al. [29], who observed a reduction in *P*_N_ after 45 days of flooding in cocoa genotypes, except for SJ-02, noting that the longer the plant stress period, the greater the reduction in P_N_ as a consequence of the inhibition of photosynthesis.

Flooding conditions caused a reduction in *g*_s_ in all rootstocks (Figure 3). However, SJ-02 did not show any difference in its behavior between environments when compared to the other genotypes (Figure 3). Reduced *g*_s_ values can be associated with degradation of the root system, since the roots are unable to maintain water absorption in the anoxic environment [5]. Thus, the lower the stomatal conductance generated by the increased stress time, the greater the loss in photosynthesis, since the plants are unable to produce ATP and NADPH2 and begin to dissipate the absorbed light energy, increasing the chances of photoinhibition [30].

It is noted that in our study, there was a reduction in transpiration after the flooded environment compared to the control. Bertolde et al. [13] found an 80% reduction in transpiration compared to the control after 20 days of flooding, corroborating the data found in this work. However, it should be highlighted that the SJ-02 genotype maintained transpiration levels in the flooded environment similar to the control, suggesting greater plasticity to this environment.

When stomatal limitations occur, there is consequently a reduction in CO_2_ capture, and, thus, a reduction in the *C*_i_/*C*_a_ ratio, this reduction being related to the reduction in *g*_s_. However, it should be noted that all rootstocks presented *C*_i_/*C*_a_ values close to or greater than 0.7 (Figure 4), which, according to [13], is indicative that stomatal restriction was not sufficient to limit the *C*_i_/*C*_a_ ratio [31].

The stress generated by flooding conditions induced a reduction in water-use efficiency in all genotypes, with the most pronounced reductions in the TSH-1188 and Esfip-02 genotypes (Figure 5). Unlike the data observed in our study, for flooded water conditions, when plants present high values of WUE, it is indicative of greater physiological plasticity, given the stressful conditions of the environment [32,33]. Thus, the increase in water-use efficiency during flooding indicates that the plant is using more water in this environment [34]. Therefore, it is possible to verify with the data presented in our study that the cocoa plants had a reduced water absorption capacity by the roots.

In addition to stomatal closure, chlorophyll degradation and the accumulation of soluble sugar in the leaves during flooding contributed to the reduction in photosynthesis [35]. High values of the SPAD index indicate that the FSII is not compromised [5], but this did not occur with the rootstocks studied, since all showed a decline in their green index (Figure 5). Possibly associated with the increase in acetaldehyde and acetic acid and the action of ethylene [36] in the leaves through chlorophyll degradation [37].

After stress suspension, all rootstocks reestablished *P*_N_ (Figure 6), *E* (Figure 7), and WUE (Figure 8). This is an indication that the plants managed to reestablish water absorption and reactivate the photosynthesis process. Corroborating the results presented in this study, Sena Gomes and Kozlowski [15] observed the recovery of stomatal conductance and transpiration to control levels after the suspension of flooding stress in the Catongo genotype. Almeida et al. [16] found that cocoa plants flooded for 35 days recovered 78 to 100% of their photosynthetic capacity 10 days after the flooding was lifted. Pinto et al. [14] reported impaired photosynthetic function in the genotypes TSH-1188, Esfip-02, Cepec-2002, SJ-02, and Pará during flooding, with normal function restored after moisture normalization. Possibly, the recovery in gas exchange can be associated with better management of reactive oxygen compounds, improving photosynthetic acclimation after flooding [8].

### 4.2. Adaptations of the Anatomy of the Stem and Leaf

All rootstocks showed lenticel formation (Table 2, Figure 9), with no differences between them under flooding conditions. The appearance of lenticels is quite common in cocoa genotypes as a form of adaptation to stress; in our study, all rootstocks showed lenticel formation nine days after flooding, with no differences between them under flooding conditions. In different studies, the appearance of lenticels on the stem of cocoa has been observed during flooding, after 3, 8, 10, and 12 days of the plants being under this condition [3,13,38,39]. In our study, all rootstocks showed the formation of lenticels after nine days of flooding, with no differences between them in the flooded condition. The length of the lenticels remained the same during recovery, except for the SJ-02 and Pará genotypes, and their height regressed when the plants were removed from the flooding and placed to recover (Table 2).

Lenticels are structures that facilitate the transport of oxygen to the roots and protect the tissues from toxic substances arising from fermentation [40]. The presence of hypertrophied lenticels promotes the release of toxic compounds that are formed by the roots [41], such as the production of lactate, acetaldehyde, and ethanol [42]. Among these compounds, the most harmful during flooding is ethanol. When the plant is unable to eliminate all the ethanol produced, it causes darkening of the root system and consequent death, impairing the growth of new roots [5].

Based on the data shown, it can be seen that the length of the lenticels remained the same during recovery, except for the SJ-02 and Pará genotypes, and their height decreased when the plants were removed from the flooded area and placed to recover (Table 2). Kolb et al. [43], working with flooded plants, observed that when comparing the height and length of the lenticels with those of plants grown in drained soils, there were significant differences between the environments. The presence of lenticels indicates that oxygen is not completely absent, allowing plants to survive in this type of environment [44].

The increase in stomatal density in the SJ-02, Pará, and Esfip-02 genotypes during flooding contributed to the conservation of CO_2_ assimilation, being an essential strategy to increase photosynthesis and tolerate stress conditions [45], as observed for these genotypes during the recovery process. On the other hand, the TSH-1188 and Cepec-2002 rootstocks showed a reduction in stomatal density during flooding, associated with stomatal closure. This response can be considered to be a form of protection by plants to avoid excessive water loss [46], becoming a survival mechanism against water saturation of the root system [30].

The return of stomatal density similar to the non-flooded environment of the genotypes after the suspension of flooding stress confirms that changes in the density and size of stomata in leaves developed in flooded environments are a sign of the plasticity of the species to adapt to the environment [43]. The Pará rootstock did not present statistical differences between the three environments (Table 2), showing that it was the only one that did not present plasticity.

### 4.3. Pigment Contents

Flooding negatively influenced pigments, reducing the plants’ light absorption capacity. Chlorophyll and carotenoids are directly linked to the photosynthetic capacity of plants, as they have functions in the absorption and transfer of light energy to the PSII [47], and are essential for plant development [48]. The stressful condition generated by excess water and reduced oxygen concentration in the roots limits the presence of nitrogen and magnesium in the leaves, which are constituents of the pigments [49].

The reduction in chlorophyll is due to an increase in the chlorophyllase enzyme, which degrades its molecules. This is because stress inhibits the chlorophyll precursor molecule 5-aminolevulinic acid [50]. Bertold et al. [13], studying cocoa clones, found a reduction in chlorophyll *a* and *b* 30 days after flooding, a response similar to that found in the present work, where, after 60 days of flooding, chlorophyll degradation caused by flooding had occurred (Table 3). Chlorophyll b is responsible for capturing light by transferring it to chlorophyll a and thus participating in photochemical reactions [50]. With the reduction in this chlorophyll in the flooded environment (Table 3), it can be said that there was a reduction in light capture. Flooding favored a reduction in the a/b ratio (Table 3), indicating a disruption in the functioning of the photochemical phase. Furthermore, reductions in chlorophyll levels favor the appearance of reactive oxygen species that oxidize the pigments [51].

Analyzing the environments, it was found that flooding degraded carotenoids. The Cepec-2002 and Esfip-02 genotypes showed higher carotenoid values compared to the Pará and TSH-1188 genotypes (Table 3). Carotenoids are responsible for protecting photosystems against photodegradation and photooxidation [52]. The reduction in this pigment confirms that the cocoa plants suffered photodegradation caused by the long period of flooding and consequent photosynthetic decline [53].

### 4.4. The Carbohydrates

Flooding stress on the root system generates changes in plants due to the lack of oxygen in the roots, leading the plant to adapt, such as through the accumulation of metabolites in the aerial part [28], since the plant tends to save energy to ensure its survival, thus tending to maintain a balance in the distribution of photoassimilates until the flooding is suspended [54]. This fact explains the accumulation of reducing sugar (RS) in the leaves of plants kept under flooding, with emphasis on the TSH-1188 genotype (Table 4). It is also known that plants under flooding store carbohydrates as a way to prevent oxidative stress [55].

The data obtained in this study also indicate that there was an accumulation of total soluble sugar (TSS) in the leaves of the genotypes in the flooded environment, with higher TSS in the leaves of the Esfip-02 and Cepec-2002 genotypes (Table 4). The increase in soluble sugars in the leaves contributed to the osmoregulation of the plants. This accumulation in the leaves probably reduces A rates and contributes to a satisfactory pressure potential and maintains the water content inside the cells, thus avoiding and reducing the damage caused by stress [34,35,36,37,38,39,40,41,42,43,44,45,46,47,48,49,50,51,52,53,54,55,56]. It should be noted that the TSS produced tends to accumulate in the aerial part of the plants, as there is less translocation from the leaf to the roots due to the compromise of the phloem, which can cause damage to photosynthetic rates [57].

The allocation of carbohydrates in the root system showed differences between environments, where flooding favored an increase in TSS in all rootstocks (Table 4). The increased accumulation of carbohydrates in the TSS root system observed in plants under flooding, as presented in our study, is reported as a survival mechanism in environments with flooding stress [22,23,24,25,26,27,28,29,30,31,32,33,34,35,36,37,38,39,40,41,42,43,44,45,46,47,48,49,50,51,52,53,54,55,56,57,58]. Thus, under this type of stress, plants were unable to utilize carbohydrates, which is a sign of low tolerance to flooding [35]. Thus, this adaptation allows the plant to modify its gene expression, responsible for conferring tolerance to stressful conditions [59].

Under flooding conditions, the lower rate of phloem transport to the roots results in starch accumulation, as observed in the leaves and roots of plants when compared to the non-flooded environment, while the Cepec-2002 and Pará rootstocks showed lower starch accumulations (Table 4). This occurs because the root system under anoxia requires large amounts of sugars for anaerobic respiration [29]. The maintenance of glycolysis and the induction of fermentation during flooding are potential characteristics that confer tolerance to flooding [28]. Since carbohydrate formation depends on the efficient capture of light energy and oxygen supplies, the reduction of oxygen caused by flooding hinders photosynthesis and respiration, consequently hindering carbohydrate formation [60].

The data obtained demonstrate that, during recovery from stress, there was a reduction in RS, TSS, and starch in the leaves, with no differences between the rootstocks (Table 4). The root system during the recovery period generated an increase in starch and a reduction in TSS (Table 4). This fact is related to the increased biosynthesis through the use of stored carbohydrates; thus, the plants suffered less stress during recovery [61]. It should be noted that the plant recovery phase after flooding can be equally stressful, being related to plant performance during this period [8]. Thus, plant survival after flooding may be related to the ability to maintain carbohydrate levels [62].

It is worth noting that the adaptation of cocoa plants to the flooded environment is related to a series of factors that allow the plants to survive in stressful and non-ideal conditions for their development. Furthermore, the morphophysiological modifications of the plants, such as the appearance of lenticels, stomatal density, and carbohydrate and photosynthetic pigment contents, are essential for the adaptive success of the species.

Thus, we highlight that the use of the genotypes TSH-1188, Cepec-2002, Pará, Esfip-02, and SJ-02 in our study is due to the need to characterize these genotypes in flooded environments. These genotypes are commonly used as rootstocks in cocoa production; however, there are few studies characterizing these genotypes regarding their tolerance to flooding and the recovery of plants after suffering stress. Furthermore, the differences observed in the genotypes are possibly due to their original and parental characteristics. However, as demonstrated, although the plants suffered alterations in photosynthesis during flooding, all genotypes resumed their normal functions after recovering from the stress, demonstrating that they may be suitable for planting in locations subject to flooding. This finding is a milestone for the production of seedlings through grafting, as it allows farmers to plan crop management and assists in decision-making regarding the genotypes that should be planted in their area.

Furthermore, since cocoa cultivation is a dynamic activity requiring periodic crop renewal, studies characterizing other genotypes with potential for planting in areas with biotic and abiotic limitations, such as tolerance to pests and diseases and thermal, light, water, and saline stress, are fundamental, potentially ensuring greater sustainability for the crop.

## 5. Conclusions

The exposure time of the cocoa (*Theobroma cacao* L.) rootstocks TSH-1188, Cepec-2002, Pará, Esfip-02, and SJ-02 to flooding caused limitations in gas exchange. At 55 days after flooding, all genotypes showed reductions in net photosynthetic assimilation, stomatal conductance, and transpiration rate in the flooded environment compared to the non-flooded environment.

The photosynthetic pigments of the cocoa rootstocks TSH-1188, Cepec-2002, Pará, Esfip-02, and SJ-02 were degraded during flooding, with average values of Chl *a*, Chl *b*, total Chl, and total carotenoids of 9.33, 10.418, 19.75, and 590.75 μg.mL^−1^ in the non-flooded environment and 6.43, 7.69, 14.12, and 500.33 μg.mL^−1^ in the flooded environment. The rootstocks Cepec-2002 and Esfip-02 showed higher carotenoid accumulation with 585.78 and 650.47 μg.mL^−1^, respectively, when compared with SJ-02 (474.03 μg.mL^−1^), Pará (491.58 μg.mL^−1^), and TSH-1188 (525.86, 491.58 μg.mL^−1^).

The Pará rootstock did not show differences in stomatal density between environments, with values of 32.25 in flooding, 34.83 in non-flooding, and 31.61 in recovery.

After nine days of flooding, the cocoa genotypes TSH-1188, Cepec-2002, Pará, Esfip-02, and SJ-02 showed morphological changes through the appearance of hypertrophied lenticels on the stem, suggesting adaptation to stress.

After recovery, the rootstocks TSH-1188, Cepec-2002, Pará, Esfip-02, and SJ-02 showed the ability to normalize their gas exchange, carbohydrate exchange, and anatomical processes, demonstrating that the root system managed to re-establish its functions and indicating tolerance of these genotypes to areas at risk of flooding.

## Figures and Tables

**Figure 1 plants-15-00122-f001:**
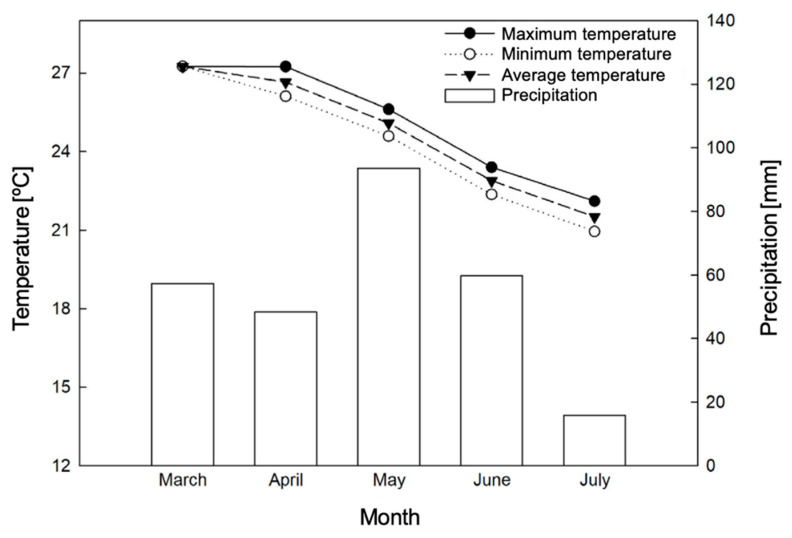
Total precipitation [mm], relative humidity [%], and maximum [°C], average [°C], and minimum temperatures [°C] recorded at the Linhares weather station (ES), during the acclimatization process and evaluation of the experiment.

**Figure 2 plants-15-00122-f002:**
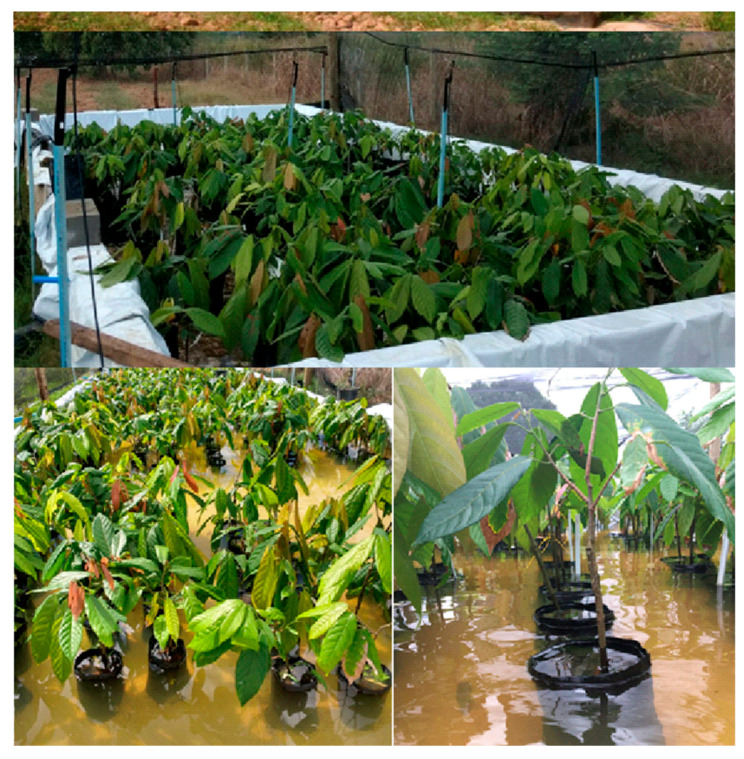
General view of the cocoa seedling flooding experiment. Masonry tanks lined with white tarpaulin to prevent water infiltration, used in conducting the experiment.

**Figure 3 plants-15-00122-f003:**
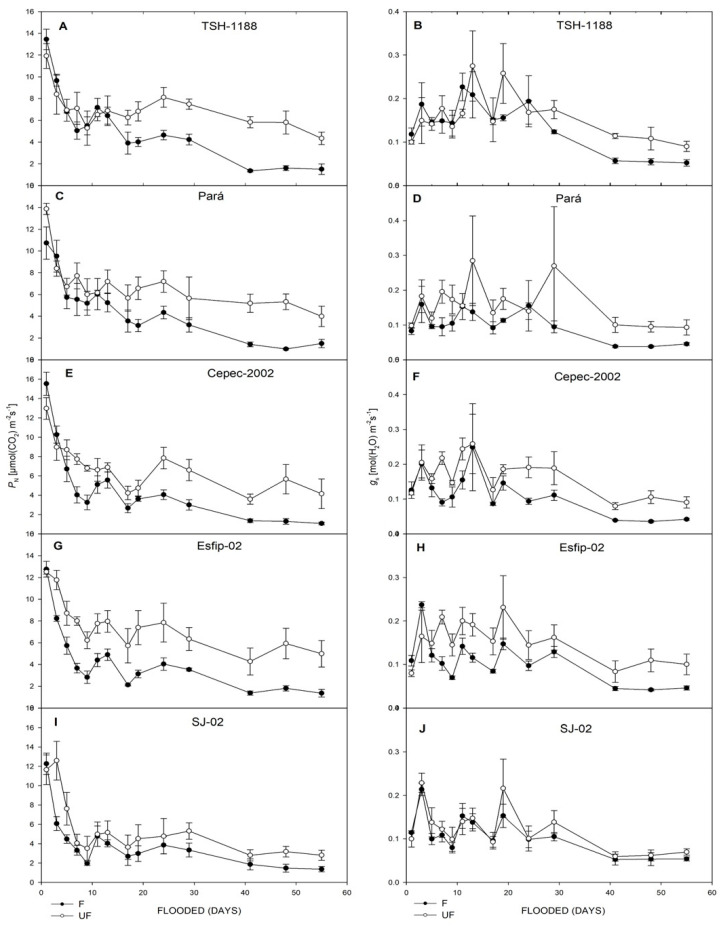
Net photosynthetic rate (*P*_N_) (**A**,**C**,**E**,**G**,**I**), and *g*_s_ stomatal conductance (**B**,**D**,**F**,**H**,**J**), of unflooded (UF) and flooded (F) cocoa genotypes TSH-1188, Pará, Cepec-2002, Esfip-02, and SJ-02, evaluated over 55 days of flooding.

**Figure 4 plants-15-00122-f004:**
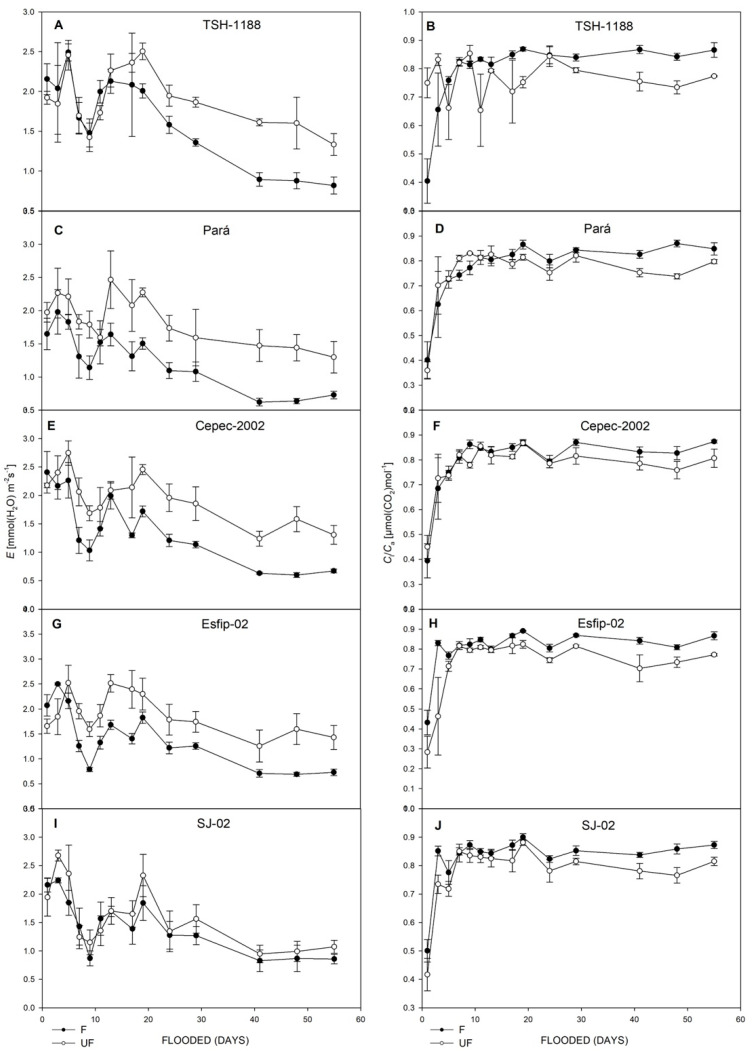
Transpiration (*E*) (**A**,**C**,**E**,**G**,**I**), and ratio of intra/extracellular CO_2_ concentrations (*C*_i_/*C*_a_), (**B**,**D**,**F**,**H**,**J**), of unflooded (UF) and flooded (F) cocoa genotypes TSH-1188, Pará, Cepec-2002, Esfip-02, and SJ-02, evaluated over 55 days of flooding.

**Figure 5 plants-15-00122-f005:**
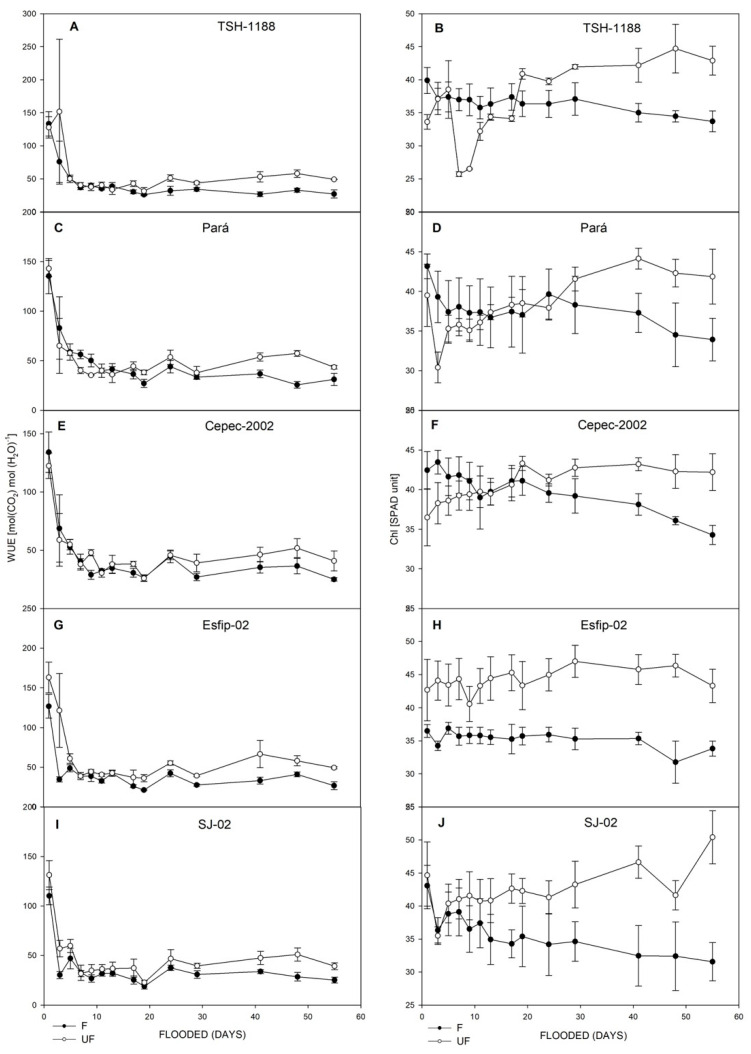
Water-use efficiency (WUE) (**A**,**C**,**E**,**G**,**I**), and Chl (SPAD unit) (**B**,**D**,**F**,**H**,**J**), of unflooded (UF) and flooded (F) cocoa genotypes TSH-1188, Pará, Cepec-2002, Esfip-02, and SJ-02, evaluated over 55 days of flooding.

**Figure 6 plants-15-00122-f006:**
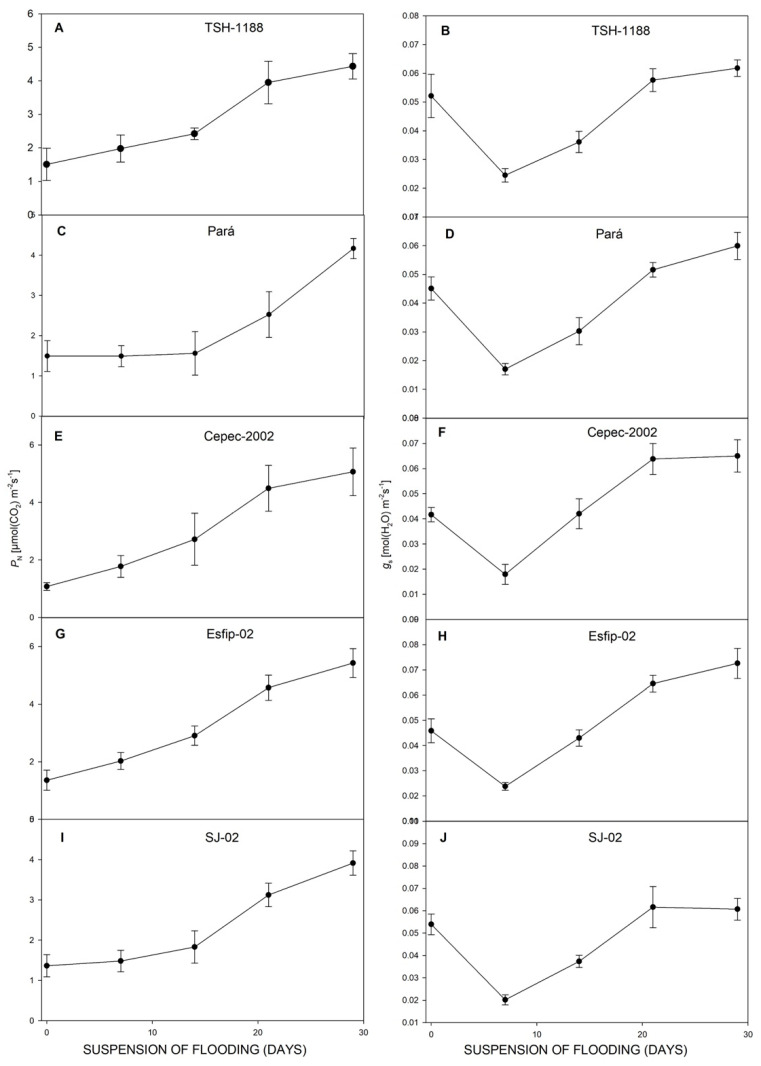
Net photosynthetic rate (*P*_N_) (**A**,**C**,**E**,**G**,**I**), and *g*_s_ stomatal conductance (**B**,**D**,**F**,**H**,**J**), of cocoa genotypes TSH-1188, Pará, Cepec-2002, Esfip-02, and SJ-02 during recovery after the flooding interruption. The starting point represents the last assessment (55 days of flooding).

**Figure 7 plants-15-00122-f007:**
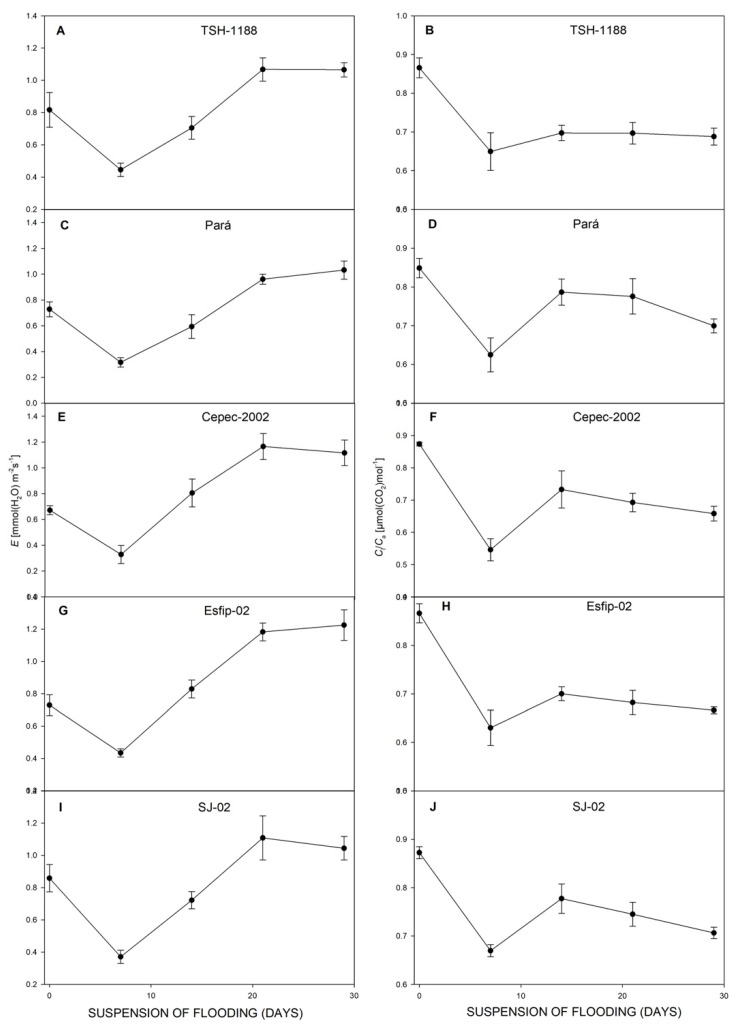
Transpiration (*E*) (**A**,**C**,**E**,**G**,**I**), and ratio of intra/extracellular CO_2_ concentrations (*C*_i_/*C*_a_), (**B**,**D**,**F**,**H**,**J**), of cocoa genotypes TSH-1188, Pará, Cepec-2002, Esfip-02, and SJ-02 during recovery after the flooding interruption. The starting point represents the last assessment (55 days of flooding).

**Figure 8 plants-15-00122-f008:**
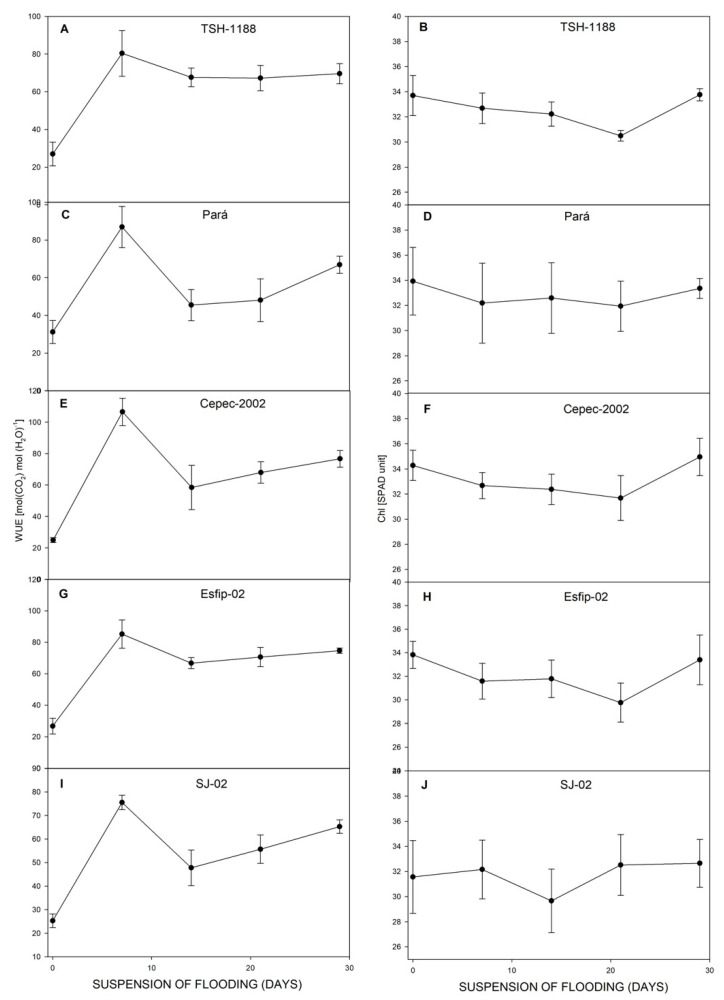
Water-use efficiency (WUE) (**A**,**C**,**E**,**G**,**I**), and Chl (SPAD unit) (**B**,**D**,**F**,**H**,**J**), of cocoa genotypes TSH-1188, Pará, Cepec-2002, Esfip-02, and SJ-02 during recovery after the flooding interruption. The starting point represents the last assessment (55 days of flooding).

**Figure 9 plants-15-00122-f009:**
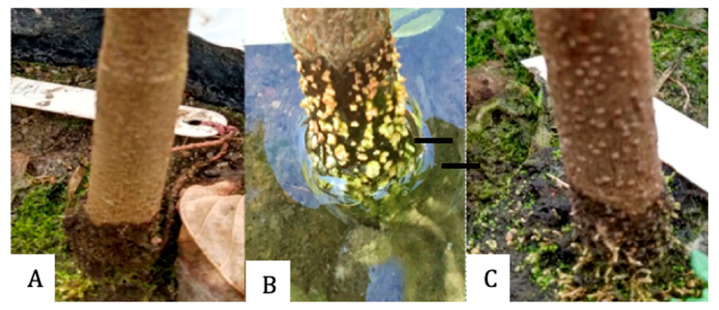
Formation of hypertrophied lenticels in the stem during flooding stress of *Theobroma cacao* L. seedlings: (**A**) non-flooded stem, (**B**) flooded stem, and (**C**) stem after recovery.

**Figure 10 plants-15-00122-f010:**
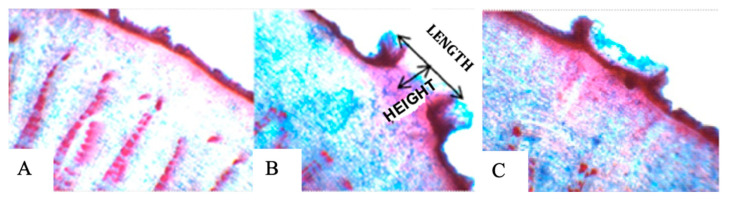
Anatomical sections of hypertrophied lenticels on the stem during flooding stress of *Theobroma cacao* L. seedlings: (**A**) non-flooded stem, (**B**) flooded stem, and (**C**) stem after recovery.

**Table 1 plants-15-00122-t001:** Genealogy and agronomic descriptors containing the origin, parents, ancestry, type of pollination, and fruit color and shape of the cocoa (*Theobroma cacao* L.) genotypes Pará, TSH-1188, Esfip-02, Cepec-2002, SJ-02, and PS-1319.

Genotype	Origin	Parent	Ancestry	Pollination	Fruit Color and Formation
Pará	Bahia	Undefined	Forastero	SC	Y/AM
TSH-1188	Trinidad and Tobago	IMC67, ICS1, SCA6, and P18	Amazônico/Trinitário	SI	R/EL
Esfip-02	Espírito Santo Region	TSH-565 and IMC-67	Trinitário/Forastero	SI	R/AL
Cepec-2002	Brasileira Farm, Uruçuca-BA	Sca-6 and Comum **	Amazônico/Amazônico	SC	Y/AM
SJ-02	São José Farm, Itajuipe-BA	IMC-67 and ICS-01	Amazônico/Trinitário	SC	Y/AL
PS-1319	Porto Seguro Farm, Ilhéus-BA	ICS-01 and PA-150	Amazônico/Trinitário	SC	Y-V/AM

SC—Self-compatibility; SI—self-incompatibility; Y—yellow; R—red; AM—amelonado; EL—elongated. ** Catongo in its composition.

**Table 2 plants-15-00122-t002:** Leaf and stem anatomical characteristics (stomatal density, polar diameter/equatorial diameter ratio (DP/DE ratio), lenticel length, and lenticel height), evaluated in flooded and non-flooded plants after 60 days of flooding and recovered after 30 days of flooding suspension in the five assessed cocoa genotypes.

	Genotypes					
Treatment	Cepec-2002	SJ-02	Esfip-02	Pará	TSH-1188	Means
Stomatal density						
Non-flooded	35.96 bA	33.70 bC	35.16 bB	34.83 bA	40.48 aA	
Flooded	25.16 cB	42.74 aA	43.38 aA	32.25 bA	29.35 bC	
Recovered	33.54 bA	37.58 aB	32.74 bB	31.61 bA	35.64 aB	
DP/DE ratio						
Non-flooded	0.92 aA	1.0 aA	1.0 aA	1.05 aA	0.97 aA	
Flooded	1.07 aA	0.75 bB	0.97 aA	0.95 aA	0.85 bA	
Recovered	0.97 aA	0.95 aA	1.0 aA	1.0 aA	0.97 aA	
Lenticel length						
Non-flooded	0 aB	0 aC	0 aB	0 aC	0 aB	
Flooded	0.55 aA	0.65 aA	0.63 aA	0.61 aA	0.58 aA	
Recovered	0.54 bA	0.50 bB	0.67 aA	0.50 bB	0.58 bA	
Lenticel height						
Non-flooded	0	0	0	0	0	0 C
Flooded	0.55	0.65	0.63	0.61	0.58	0.22 A
Recovered	0.54	0.5	0.67	0.5	0.58	0.15 B

Means followed by the same letter are not different from each other by the Scott–Knott test, at 5% probability. Upper-case letters are used to compare environments in the columns, and the lower-case letters compare genotypes in the rows.

**Table 3 plants-15-00122-t003:** Analysis of Chlorophyll (Chl *a*, Chl *b,* and Chl total) [μg mL^−1^] and Total carotenoids [μg mL^−1^] in flooded and non-flooded environments evaluated after 60 days of flooding.

	Genotypes					
Environment	Cepec-2002	SJ-02	Esfip-02	Pará	TSH-1188	Means
Chl *a* [μg mL^−1^]						
Non-flooded	9.12	10.24	11.15	9.27	6.9	9.33 A
Flooded	7.46	5.7	5.85	6.58	6.57	6.43 B
Chl *b* [μg mL^−1^]						
Non-flooded	10.0925	11.57	13.14	9.8725	7.415	10.418 A
Flooded	9.5125	6.0025	6.8125	7.6975	8.425	7.69 B
Chl total [μg mL^−1^]						
Non-flooded	19.2	21.8	24.28	19.14	14.31	19.75 A
Flooded	16.96	11.7	12.66	14.28	14.99	14.12 B
Total carotenoids [μg mL^−1^]						
Non-flooded	619.68	555.83	684.9	513.76	579.59	590.75 A
Flooded	551.88	392.23	616.04	469.39	472.13	500.33 B
Means	585.78 a	474.03 b	650.47 a	491.58 b	525.86 b	

Means followed by the same letter are not different from each other by the Scott–Knott test, at 5% probability. Upper-case letters are used to compare environments in the columns, and the lower-case letters compare genotypes in the rows.

**Table 4 plants-15-00122-t004:** Quantification of carbohydrates reducing sugar (RS) [mg g^−1^] DM of leaf, total soluble sugar (TSS) [mg g^−1^] DM of leaf, and root of five genotypes of non-flooded, flooded, and recovered cocoa.

	Genotype					
Environment	Esfip-02	Cepec-2002	SJ-02	Pará	TSH-1188	Means
RS Leaf [mg g^−1^]						
Non-flooded	43.40 aC	42.51 aC	38.11 aB	37.91 aC	45.65 aC	
Flooded	77.42 bA	77.75 bA	58.42 cA	78.49 bA	101.12 aA	
Recovered	60.36 aB	63.05 Ab	57.23 aA	53.49 aB	65.62 aB	
TSS Leaf [mg g^−1^]						
Non-flooded	60.83 aB	54.33 aB	52.33 aB	50.73 aC	70.42 aB	
Flooded	134.47 aA	137.43 aA	115.26 bA	108.14 bA	98.43 bA	
Recovered	61.61 aB	73.63 aB	65.24 aB	79.98 aB	69.80 aB	
Starch Leaf						
Non-flooded	120.99	127.13	127.98	120.63	119.97	123.34 C
Flooded	150.54	155.7	157.58	144.74	147.82	151.28 A
Recovered	138.13	139.79	137.18	144.42	147.76	141.46 B
Starch Root						
Non-flooded	143.98	130.15	152.65	132.14	150.3	141.84 C
Flooded	182.01	124.48	188.29	156.26	190.8	168.36 B
Recovered	211.94	167.66	187.02	178.63	197.31	188.51 A
Means	179.31 a	140.76 b	175.99 a	155.67 b	179.47 a	
TSS Root						
Non-flooded	52.26	27.32	37.5	31.27	32.55	36.18 C
Flooded	104.09	95.96	70.84	89.87	72.87	86.72 A
Recovered	83.31	42.02	81.08	56.23	71.66	67.46 B

Means followed by the same letter are not different from each other by the Scott–Knott test, at 5% probability. Upper-case letters are used to compare environments in the columns, and the lower-case letters compare genotypes in the rows.

## Data Availability

The original contributions presented in this study are included in the article. Further inquiries can be directed to the corresponding authors.

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
