# Peer review of "Physiological and Morphological Response Mechanisms of Theobroma cacao L. Rootstocks Under Flooding and Evaluation of Their Adaptability"

_plants, 2026, doi:10.3390/plants15010122_

Round 1

Reviewer 1 Report

Comments and Suggestions for Authors

The objective of this study was to evaluate the physiological, biochemical and anatomical changes in response to flooding and the recovery capacity of Theobroma cacao L. rootstocks. However, this paper only systematically analyzed the adaptation mode and tolerance ability of cocoa seedlings under flooding, and did not clearly rank the tolerance ability of different rootstocks. Therefore, it is suggested to add a review on the impact of rootstock on the adaptability of grafted seedlings, especially on the study of flooding, in the Introduction. Add the ranking of rootstocks in terms of water resistance in the Results, and also include the impact of rootstocks on the adaptability of grafted seedlings in the Discussion.

The specific opinions are as follows:

  1. Suggest modifying the title to "Physiological and Morphological Response Mechanisms of Theobroma cacao Rootstocks Under flooding and Evaluation of Their Adaptability".
  2. Abstract: It is suggested to reorganize and clarify the physiological and morphological mechanisms of cocoa seedlings adapting to flooding, as well as the adaptability of different rootstocks.
  3. Introduction: Add a review on the effects of rootstock on the adaptation of grafted seedlings after the third paragraph.
  4. Materials and Methods
  • Add background information on the sources of scions and rootstocks in the second paragraph of section 2.1.
  • On line 94, please provide the duration of each irrigation session.
  • In the fourth paragraph of section 2.1, provide information on the individual size of 9-month-old seedlings. What is the depth of flooding in the experiment?
  • How to determine the saturation intensity on line 119?
  • Supplement the specific sampling time in 2.3. At the same time, supplement the units of the measured indicators.
  • In 2.6, clearly define the analysis of variance method adopted for each indicator.
  1. Results

    In 3.1, supplement whether the differences among different rootstocks in terms of indicators are significant and the specific ranking of the rootstocks.

    It is suggested to increase the correlation analysis among indicators and use mathematical methods to clarify the ranking of the adaptability of rootstocks.

  1. Discussion

    It is suggested that the relevant analysis results be applied to the discussion, and at the same time, the content of comparing the adaptability of rootstocks submerged in water should be added.

  1. Conclusion

It is suggested to reorganize and clarify the main physiological mechanisms of cocoa adaptation to flooding and the ranking of adaptability between different rootstocks, in order to explore the significance and application prospects of this research result.

  1. The format of the references is not uniform. Please carefully check and modify them, such as Article 3, Article 19, Article 21, etc.
  2. In line 153, when citing references [17] and [18], it is recommended to write the author's name instead of using the reference number.
  3. In the main text, use the full name for the first Latin name and the abbreviation for the subsequent ones. Please make careful modifications.

Reviewer 2 Report

Comments and Suggestions for Authors

This topic is very interesting and relevant. The scientific world needs such research. However, there are very serious shortcomings in the presentation of this study:
1. The abstract needs to be clarified with specific results and figures.
2. The introductory chapter itself lacks any information on this topic for Theobroma cacao L. Numerous articles have been published on the topic of the effect of flooding stress on cocoa ( Theobroma cacao L.). Therefore, it is necessary to thoroughly analyze this topic and understand the novelty of your specific research. The introduction is still extremely underdeveloped.
3. Why is there no error bar in Figure 1? The fluctuations over the course of a month are unclear.
4. Why are some methods unlinked and unnamed? How much did the pots weigh before the experiment?
5. There is no detailed description of the genotypes and how they differ.
6. How many leaves and from how many plants of each genotype were used in each experiment?
7. The Discussion also provides little comparison with previously published studies on this topic, specifically on cocoa. This chapter largely duplicates the description of the results, without providing a full analysis. Since the genotype characteristics are not described, it's unclear why the results differ. It would be possible to speculate about why the data differs, and why the stress management strategies differ.
8. The data are quite good, but very poorly discussed.
9. The Conclusions section is also simply a statement of facts without any real conclusions drawn from the analysis. There are no supporting figures, comparisons, assumptions, or ideas for further use of the data.

Comments on the Quality of English Language

This topic is very interesting and relevant. The scientific world needs such research. However, there are very serious shortcomings in the presentation of this study:
1. The abstract needs to be clarified with specific results and figures.
2. The introductory chapter itself lacks any information on this topic for Theobroma cacao L. Numerous articles have been published on the topic of the effect of flooding stress on cocoa ( Theobroma cacao L.). Therefore, it is necessary to thoroughly analyze this topic and understand the novelty of your specific research. The introduction is still extremely underdeveloped.
3. Why is there no error bar in Figure 1? The fluctuations over the course of a month are unclear.
4. Why are some methods unlinked and unnamed? How much did the pots weigh before the experiment?
5. There is no detailed description of the genotypes and how they differ.
6. How many leaves and from how many plants of each genotype were used in each experiment?
7. The Discussion also provides little comparison with previously published studies on this topic, specifically on cocoa. This chapter largely duplicates the description of the results, without providing a full analysis. Since the genotype characteristics are not described, it's unclear why the results differ. It would be possible to speculate about why the data differs, and why the stress management strategies differ.
8. The data are quite good, but very poorly discussed.
9. The Conclusions section is also simply a statement of facts without any real conclusions drawn from the analysis. There are no supporting figures, comparisons, assumptions, or ideas for further use of the data.

Reviewer 3 Report

Comments and Suggestions for Authors

Manuscript Title: Effect of flooding and recovery capacity after stress suspension of Theobroma cacao L. rootstocks

Overall Comment

This study analysis a relevant and timely topic concerning the physiological, biochemical, and anatomical responses of cacao (Theobroma cacao L.) rootstocks to prolonged flooding stress and their subsequent recovery capacity. The work has real-world implications for cacao cultivation in flood-prone regions, especially in tropical production systems where climate variation is increasing the frequency and duration of waterlogging events. The study is generally well-structured and offers valuable perceptions; however, several parts require clarification, improvement, and deeper scientific interpretation to enhance rigor, reproducibility, and impact.

  1. Major Comments
  2. The Introduction provides a general explanation of flooding stress but lacks a clear articulation of the specific knowledge gaps in cacao physiology, the rationale for selecting the studied rootstocks, and a concise, hypothesis-driven statement of the study’s objectives.
  3. Please build up the scientific justification by arguing known physiological thresholds, morphological adaptations (e.g., lenticel hypertrophy), and genotype-specific flooding resilience from previous studies.
  4. The justification for selecting the five specific rootstocks (TSH-1188, Cepec-2002, Pará, Esfip-02, SJ-02) should be explained in more detail. Are these widely used in breeding programs? Known for vigor? Suspected differential tolerance?
  5. The study would benefit from more clear details of experimental replication, plant age, growth conditions (light intensity, humidity, soil texture), and sampling intervals.
  6. It rests unclear whether the flooding depth was constant and how oxygen availability was controlled/monitored.
  7. Please explain how “recovery capacity” was operationally defined what criteria determined full or partial recovery?
  8. The decrease in photosynthesis, stomatal conductance, and transpiration is expected under hypoxic stress; however, statistical comparisons among rootstocks require deeper analysis.
  9. The study should include standard error bars and clear p-value reporting for all gas-exchange parameters.
  10. Elucidation of pigment degradation is appropriate, yet mechanistic explanations (chlorophyll breakdown pathways, ROS involvement, photoinhibition) are missing.
  11. The accumulation of total soluble sugars could show osmotic adjustment or impaired phloem transport. This needs further discussion with supporting citations.
  12. Study correlating pigment loss with photosynthetic suppression to strengthen functional interpretation.
  13. Lenticel formation is well stated, but the anatomical micrographs require higher resolution and more consistent labeling.
  14. The study references “normalization of anatomy during recovery,” but this claim requires more detailed evidence were tissues fully restored or partially modified?
  15. The return to normal gas exchange and carbohydrate levels after stress release is a key contribution of this study.
  16. However, the Discussion should more deeply report whether full physiological restoration truly implies absence of long-term damage.
  17. Several figures would benefit from clearer axis labels, uniform scales, and higher resolution.
  18. Highlight significant differences among rootstocks in each figure panel using constant annotation.

Minor Comments

  1. Numerous sentences in the Introduction and Discussion would benefit from improved conciseness and biochemical precision.
  2. Improve transition sentences to create solid linkage between physiological, biochemical, and anatomical findings.
  3. Some references are outdated; consider adding more recent studies (post-2018) on flooding stress in perennial crops.

Final Recommendation

Through moderate revision, particularly regarding experimental detail, mechanistic interpretation, statistical presentation, and figure clarity, this study has good potential for publication. The study contributes valuable knowledge to cacao stress physiology and will be useful for breeding and management strategies in flood-prone regions.

Comments on the Quality of English Language

The English is understandable, but many sentences could be made more concise and precise. In addition, transitions between sentences and paragraph would improve the flow of the study. I recommend language editing will improve clarity, coherence and overall readability.

Reviewer 4 Report

Comments and Suggestions for Authors

The study addresses an important topic regarding the effect of flooding and post-recovery of Theobroma cacao L. rootstocks, and the experimental work appears to be sound. However, the manuscript is in a preliminary stage and requires substantial revision. The writing, in particular, needs significant improvement for clarity and coherence, for example, with the title being somewhat difficult to understand and the abstract lacking a clear structure—introducing the research question and background, identifying the knowledge gap, and presenting the results in a more organized manner. The introduction section is immature and does not adequately lay the research foundation or explain why the authors focused on five cocoa rootstocks. Additionally, the materials and methods section lacks critical details, such as sample size and treatment conditions, making it difficult to assess the scientific robustness of the study. Overall, the study seems well-conducted, but the manuscript needs substantial revisions in terms of language, structure, and methodological transparency before it can be considered for publication.

Round 2

Reviewer 1 Report

Comments and Suggestions for Authors

    This manuscript has undergone numerous meaningful revisions, but still no clear ranking of the flooding tolerance of different rootstocks has been established. Clearly defining the flooding tolerance of different rootstocks is one of the important conclusions of this paper. Therefore, it is suggested to add the ranking of the flooding tolerance of each rootstock in the Abstract and Conclusion.

Specific suggestions are as follows:

(1) In the Materials and Methods section, it is recommended to add the time of the experiments.

(2) For the vertical axis, horizontal axis and symbol explanations in Figure 1, it is suggested that the first letter of each word should be capitalized, and the following letters should be in lowercase.

(3) On line 170, it is recommended to add how the saturated light intensity was determined.

(4) For Tables 1, 2, 3 and 4, it is recommended to use three-line tables.

(5) In line 358, the Latin names should use abbreviations.

(6) It is suggested to reorganize the conclusion. First summarize the flooding tolerance mechanism of cocoa, then clearly define the differences in flooding tolerance among different rootstocks, and finally look forward to the significance and application prospects of this research result.

Reviewer 2 Report

Comments and Suggestions for Authors

Thank you for your excellent work on the manuscript. Much has become clearer. However, there are still several very important points:
1. The abstract should clearly indicate the phenotypic and physiological differences between the rootstocks. Why the difference? Again, the interpretation of the results is unclear. For example, "The Cepec-2002 and Esfip-02 rootstocks showed greater accumulation of carotenoids." By what percentage? For example, 2% is still an increase, but it is not statistically significant. This is also true throughout the abstract and conclusions.
2. And yet, in the Introduction, the importance of this particular study, why these particular genotypes, and what questions remain unanswered are still unresolved. Why is this study necessary at all?
3. Figure 9 illustrates the results very well, but the quality of the images themselves is poor. Perhaps there is a photo where the plants themselves are in focus?
4. The manuscript still doesn't explain why so many genotypes were sampled, why the results were so different for each genotype, or what the possible cause might be. The analysis has improved, but not in this regard.

Comments on the Quality of English Language

Thank you for your excellent work on the manuscript. Much has become clearer. However, there are still several very important points:
1. The abstract should clearly indicate the phenotypic and physiological differences between the rootstocks. Why the difference? Again, the interpretation of the results is unclear. For example, "The Cepec-2002 and Esfip-02 rootstocks showed greater accumulation of carotenoids." By what percentage? For example, 2% is still an increase, but it is not statistically significant. This is also true throughout the abstract and conclusions.
2. And yet, in the Introduction, the importance of this particular study, why these particular genotypes, and what questions remain unanswered are still unresolved. Why is this study necessary at all?
3. Figure 9 illustrates the results very well, but the quality of the images themselves is poor. Perhaps there is a photo where the plants themselves are in focus?
4. The manuscript still doesn't explain why so many genotypes were sampled, why the results were so different for each genotype, or what the possible cause might be. The analysis has improved, but not in this regard.

Reviewer 4 Report

Comments and Suggestions for Authors

The authors appear to have made some revisions based on my previous review. However, I still feel that the core concerns have not been sufficiently addressed, and the overall writing quality remains below the expected standard. I am sorry that I cannot be more optimistic at this time...

Author Response

Dear reviewer, thank you for your feedback on the manuscript. We have tried our best to meet all your requirements. Please let us know if you have any adjustments that need to be made so that we can make the changes.

Round 3

Reviewer 2 Report

Comments and Suggestions for Authors

Thank you very much for the corrections, which greatly improved the understanding of this research. The situation regarding genotypes has become more or less clear. The article can be accepted in its current form. Among my suggestions, I would like to note that at the end of the Discussion, it would be worthwhile to offer a hypothesis about why these differences arose, especially if these genotypes are widely used, what future research plans are in this area, and what questions remain unresolved.

Comments on the Quality of English Language

Thank you very much for the corrections, which greatly improved the understanding of this research. The situation regarding genotypes has become more or less clear. The article can be accepted in its current form. Among my suggestions, I would like to note that at the end of the Discussion, it would be worthwhile to offer a hypothesis about why these differences arose, especially if these genotypes are widely used, what future research plans are in this area, and what questions remain unresolved.

Author Response

Dear reviewer,

The following paragraphs have been added to the end of the discussion as suggested:

Thus, we highlight that the use of the genotypes TSH-1188, Cepec-2002, Pará, Esfip-02, and SJ-02 in our study is due to the need to characterize these genotypes in flooded environments. These genotypes are commonly used as rootstocks in cocoa production; however, there are few studies characterizing these genotypes regarding their tolerance to flooding and the recovery of plants after suffering stress. Furthermore, the differences observed in the genotypes are possibly due to their original and parental characteristics. However, as demonstrated, although the plants suffered alterations in photosynthesis during flooding, all genotypes resumed their normal functions after recovering from the stress, demonstrating that they may be suitable for planting in locations subject to flooding. This finding is a milestone for the production of seedlings through grafting, as it allows farmers to plan crop management and assists in decision-making regarding the genotypes that should be planted in their area.

Furthermore, since cocoa cultivation is a dynamic activity requiring periodic crop renewal, studies characterizing other genotypes with potential for planting in areas with biotic and abiotic limitations, such as tolerance to pests and diseases, thermal, light, water, and saline stress, are fundamental, potentially ensuring greater sustainability for the crop.